# Association Patterns among Physical, Chemical and Microbiological Indicators of Springs in Rio Grande do Sul, Brazil

Débora Seben [1], Marcos Toebe [2], Arci D. Wastowski [3], Genésio M. da Rosa [4], Osmar D. Prestes [3], Renato Zanella [3] and Jaqueline I. Golombieski [1,*]

1   Department of Environmental Engineering and Technology, Federal University of Santa Maria (UFSM), Frederico Westphalen 98400-000, RS, Brazil
2   Department of Agronomic and Environmental Sciences, Federal University of Santa Maria (UFSM), Frederico Westphalen 98400-000, RS, Brazil
3   Department of Chemistry, Federal University of Santa Maria (UFSM), Avenida Roraima, n. 1000, Cidade Universitária, Bairro Camobi, Santa Maria 97105-900, RS, Brazil
4   Department of Forest Engineering, Federal University of Santa Maria (UFSM), Frederico Westphalen 98400-000, RS, Brazil
*   Correspondence: jaqueline.golombieski@ufsm.br

**Abstract:** This study aimed to verify the linear associations between the physical, chemical and microbiological variables of spring water. The research was developed from two seasons of spring water sample collections and evaluated physical–chemical variables such as temperature, pH, turbidity, electrical conductivity, total alkalinity, total hardness, total ammonia, nitrite, nitrate, true and apparent colors, total phosphorus, fluoride and total iron, and microbiological variables—total coliforms and *Escherichia coli*. The variables' total alkalinity, total hardness, and electrical conductivity have a strong positive correlation among them ($r > 0.50$), which is similar to what occurs with the variables' turbidity, apparent color, true color, and total iron, between nitrite and total iron, and between the turbidity and total coliforms. These correlations occur as a function of the interaction that water has with the soil and the compounds found, thus altering the quality. The springs do not have masonry protection, they only have plant protection. Moreover, this water is exposed and accessible to animals in these areas. In addition, the result of the surface and sub-surface flow effect of spring water must be considered.

**Keywords:** correlation; principal component analysis; variables; water quality

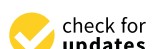



## 1. Introduction

Due to the difficulty of access and, consequently, the low pollution [1,2], groundwater represents an indispensable source for economic development and human survival [3,4]. This water is used for developing different activities such as agriculture, industry, and domestic use—including drinking water—and plays an important role in (bio)geochemical reactions in the subsurface [5–12]. This water has good stability and natural regulation, especially in drought periods [1,13,14].

Freshwater resources are very limited, and their deficiency has become a critical apprehension worldwide [11]. It is estimated that half of the world's population depends on groundwater for survival [6], as clean and safe groundwater is crucially important for social and human sustainable development. However, these resources are vulnerable to pollution due to temporal and spatial variations [14], as they interact with the soil, rocks, and the organic compounds in the environment, due to human activities [7]. Population growth and surface water contamination cause an increase in groundwater consumption [4,15]. Pollution is related to the presence of pathogens, which can cross the ground and cause the contamination of this groundwater [15].

Thus, water quality depends on the land uses', degradation processes of each watershed and the human activities developed near sources [15,16]. This research aimed to determine the correlation among the physical–chemical and microbiological variables of groundwater quality in the water used for human consumption in different land use and occupation conditions. This research is considered unprecedented and innovative because there are no other studies of these variables in these water springs.

## 2. Materials and Methods

Twenty water springs were analyzed, they are located in Rio Grande do Sul State, Brazil. The water samples were collected in two different seasons (May: autumn—collection 1; and November: spring—collection 2; 2019), using polyethylene bottles—except for the samples used for microbiological analysis, which were carried out in autoclaved glass bottles. The study area included four different land use and occupation conditions for spring water analysis, as follows: a native forest area (public area); a swine farming area (perennial condition); a soy crop area; and a tobacco crop area (annual) (five repetitions each), as detailed in [15] (Figure 1). As described in [15], in RS State, the climate is humid and sub-tropical, according to Köppen [17]. According to the Brazilian Geological Survey/CPRM(2006), geological aspects in the investigated regioncomprise the Serra Geral Formation, Paranapanema, and Lawn Facies. In the geomorphological characteristics, the water spring domains studied are located in the Rio Grande Plateau [18]. The procedure for collecting and preserving the water samples followed [19]. Five polyethylene bottles (500 mL each) were used to collect the water samples intended for physical–chemical analyses, and a glass bottle for microbiological analysis (100 mL), properly autoclaved per sampling point. All water sample analyses were carried out at the Federal University of Santa Maria (UFSM), Frederico Westphalen. The temperature and pH analyses (portable pHmeter pHEP®4 Hanna) were performed in loco. The water analyses were as follows: turbidity, electrical conductivity, total alkalinity, total hardness, total ammonia, nitrate, true and apparent colors, nitrite, total phosphorus (P), fluoride (F$^-$), total iron (Fe), total coliforms, and *Escherichia coli.* The samples were evaluated according to the methodology described in [20].

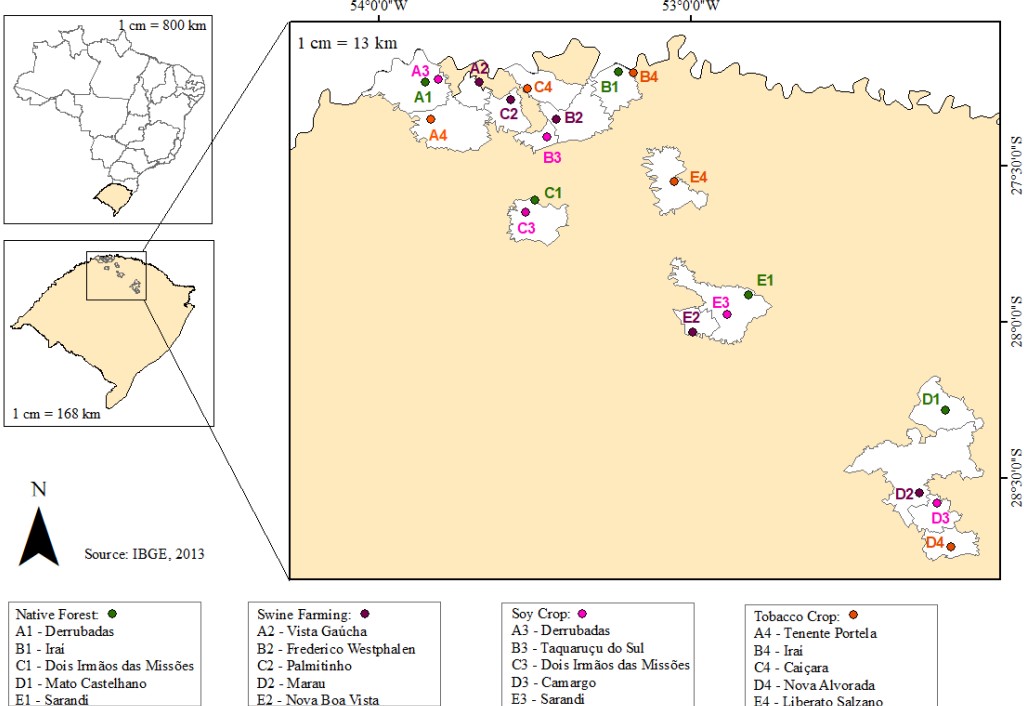

**Figure 1.** Location of the springs evaluated in each land use and occupation condition (1: native forest; 2: swine farming; 3: soy crop; and 4: tobacco crop). The cities are described in A1 to E4 [15].

Pearson linear correlation coefficients were estimated between the sixteen variables evaluated, considering the 40 sampling points. Multicollinearity diagnosis was carried out within each group of variables, through the condition number and between pairs of variables by the variance inflation factor [21], which did not indicate a serious problem of collinearity. Thus, no variable was excluded, and the canonical correlation analysis was continued to verify the existing associations between groups of variables [22]. The canonical correlations among the groups of variables were presented through the coefficients of the canonical pairs and the canonical coefficients. The significance of the canonical correlations was tested using the chi-square test ($\chi^2$) ($p \leq 0.05$). Next, based on the 16 physical–chemical and microbiological variables, the principal component analysis (PCA) was performed, according to [23]. The analyses were carried out using the R and Statistica software.

## 3. Results

The pH showed a positive correlation with the total alkalinity (TA) (r = 0.51). The TA had a positive correlation with the electrical conductivity (EC) (r = 0.85) and the total hardness (TH) (r = 0.72) (Figure 1). The TH had a positive correlation with the EC (r = 0.89) and the TA (0.72). The turbidity had a positive correlation with the total iron (TI) (r = 0.74), apparent color (AC) (r = 0.71), and total coliforms (TC) (r = 0.59). The AC showed a positive correlation with the turbidity (r = 0.71), TI (r = 0.61) and the true color (r = 0.56).

The true color showed a negative correlation with fluoride ($F^-$) (r = −0.62) and a positive correlation with the AC (r = 0.56). The TI showed a positive correlation with the nitrite ($NO_2{}^-N$) (r = 0.76), turbidity (r = 0.74), and TC (r = 0.50). $F^-$ presented a positive correlation with $NO_2{}^-N$ (r = 0.53) and a negative one with the true color (r = −0.62). The $NO_2{}^-N$ showed a positive correlation with the TI (r = 0.76) (Figure 2).

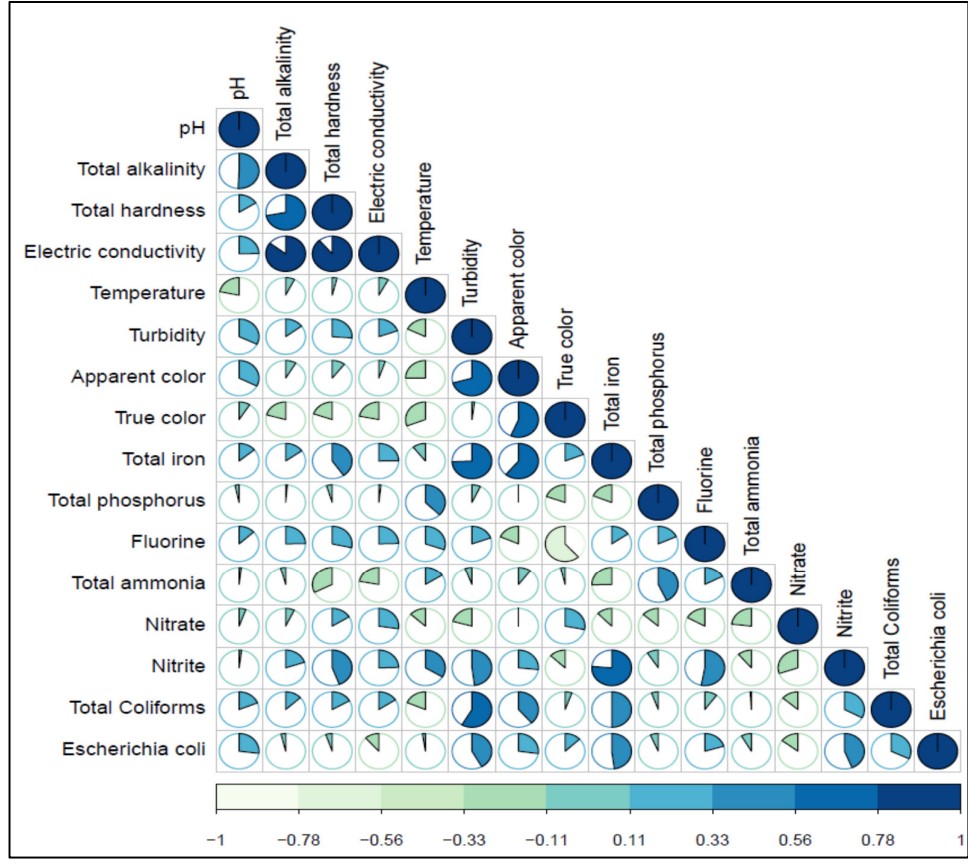

**Figure 2.** Pearson linear correlations among chemical, physical, specific chemical, and microbiological variables of springs. Variables in dark blue have a strong positive correlation and those in light green have a strong negative correlation.

Analyzing the canonical pairs, it was found that none of them were significant between the chemical and physical variables (Table 1). Assessing the coefficients of the canonical pairs between the group of chemical variables versus the specific chemical compounds, it was found that only the first canonical pair was significant. In this pair, the TI and $NO_3^-N$ had a positive linear association with the EC and a negative one with the AT, while the AM had a positive correlation with the AT and a negative one with the EC.

**Table 1.** Correlations and coefficients of estimated canonical pairs among chemical, physical, specific chemical, and microbiological variables of springs.

| Variables | Coefficients of the Canonical Pairs | | | | Variables | Coefficients of the Canonical Pairs' | |
|---|---|---|---|---|---|---|---|
| | 1st | 2nd | 3rd | 4th | | 1st | 2nd |
| Chemical | | | | | Chemical | | |
| pH | 1.18 | −0.16 | 0.3 | −0.36 | pH | 1.07 | 0.46 |
| Total Alkalinity | −1.48 | 1.32 | 1.23 | 0.49 | Total Alkalinity | −0.45 | −0.9 |
| Total Hardness | 0.81 | 0.63 | −0.02 | 1.96 | Total Hardness | 0.88 | 0.14 |
| Electric Conductivity | 0.39 | −0.88 | −1.64 | −2.38 | Electric Conductivity | −0.93 | 1.44 |
| Physical | | | | | Microbiological | | |
| Temperature | −0.37 | 0.06 | −0.43 | 0.91 | Total Coliforms | 0.08 | 1.05 |
| Turbidity | 1.03 | −0.63 | −1.42 | 0.21 | *Escherichia coli* | 0.97 | −0.41 |
| Apparent Color | −0.34 | 1.49 | 1.63 | 0.24 | | | |
| True Color | 0.38 | −1.43 | −0.45 | 0.58 | | | |
| r [1] | 0.52 ns[4] | 0.36 ns | 0.24 ns | 0.01 ns | r | 0.37 ns | 0.24 ns |
| $\chi^2$ [2] | 17.41 | 6.69 | 2.03 | 0 | $\chi^2$ | 7.18 | 2.02 |
| DF [3] | 16 | 9 | 4 | 1 | DF | 8 | 3 |
| Chemical | | | | | Microbiological | | |
| pH | 0.52 | −0.52 | 0.98 | −0.36 | Total Coliforms | 0.84 | 0.54 |
| Total Alkalinity | −1.44 | 1.21 | 0.15 | 1.46 | *Escherichia coli* | 0.78 | −0.63 |
| Total Hardness | 0.73 | 1.93 | 0.57 | −0.55 | | | |
| Electric Conductivity | 1.14 | −2.64 | −1 | 0.05 | | | |
| Chemical Species | | | | | Chemical Species | | |
| Total Iron | 0.56 | −0.64 | 1.4 | −0.46 | Total Iron | 0.96 | −0.08 |
| Total Phosphorus | 0.26 | −0.1 | −0.46 | −0.01 | Total Phosphorus | −0.12 | 0.06 |
| Fluorine | 0.34 | −0.38 | 0.79 | 0.71 | Fluorine | 0.29 | −0.52 |
| Total Ammonia | −0.5 | −0.19 | 0.29 | 0.4 | Total Ammonia | −0.09 | 0.35 |
| Nitrate | 0.55 | −0.44 | −0.53 | 0.3 | Nitrate | −0.28 | 0.09 |
| Nitrite | −0.01 | 1.31 | −1.47 | 0.36 | Nitrite | 0.74 | −0.63 |
| r | 0.71 * | 0.55 ns | 0.29 ns | 0.20 ns | R | 0.62 ns | 0.19 ns |
| $\chi^2$ | 39.63 | 16.57 | 4.33 | 1.32 | $\chi^2$ | 18.33 | 1.28 |
| DF | 24 | 15 | 8 | 3 | DF | 12 | 5 |
| Physical | | | | | Physical | | |
| Temperature | −0.21 | −0.56 | 0.87 | 0.2 | Temperature | 0.02 | −0.79 |
| Turbidity | 0.81 | −0.54 | −0.09 | 1.61 | Turbidity | 1.38 | −0.82 |
| Apparent Color | 0.11 | 0.07 | 0.29 | −2.23 | Apparent Color | −0.61 | 1.3 |
| True Color | 0.2 | 0.55 | 0.65 | 1.4 | True Color | 0.48 | −1.35 |
| Chemical Species | | | | | Microbiological | | |
| Total Iron | 1.55 | 0.18 | −0.69 | −0.32 | Total Coliforms | 0.74 | 0.75 |
| Total Phosphorus | 0.05 | −0.45 | 0.41 | 0.18 | *Escherichia coli* | 0.47 | −0.94 |
| Fluorine | 0.11 | −0.54 | −1.13 | −0.27 | | | |
| Total Ammonia | 0.14 | 0.24 | 0.36 | −1.03 | | | |
| Nitrate | −0.11 | 0.24 | 0.49 | −0.05 | | | |
| Nitrite | −0.87 | −0.45 | 1.68 | 0.45 | | | |
| r | 0.84 * | 0.76 * | 0.72 * | 0.35 ns | R | 0.66 * | 0.21 ns |
| $\chi^2$ | 98.02 | 57.54 | 29.1 | 4.48 | $\chi^2$ | 22.25 | 1.62 |
| DF | 24 | 15 | 8 | 3 | DF | 8 | 3 |

[1] r—canonical correlations. [2] $\chi^2$—chi-square test value. [3] DF—degree of freedom. [4] *—significant by the chi-square test ($p \leq 0.05$). ns—non-significant.

When the canonical pairs were compared between the group of physical variables versus the specific chemical compounds (Table 1), it was observed that, in the first canonical pair, the TI had a positive linear association with turbidity. In the second canonical pair, it was observed that P, $F^-$ and $NO_2^-N$ had a positive linear association with the temperature and turbidity, and a negative association one with the true color. On the other hand, in

the third canonical pair, the temperature had a negative linear association with $F^-$ and a positive one with $NO_2^-N$, which was maintained throughout the three canonical pairs.

The canonical correlation between the microbiological and chemical variables resulted in no significant canonical pairing (Table 1). Concerning the canonical correlation pairs between the microbiological versus physical variables, it was observed that the TC had a positive linear association with the turbidity and a negative one with the AC. The canonical correlation between specific microbiological and chemical variables also resulted in no significant canonical pairs.

The principal component analysis indicated that the first three components explained 58.88% of the total variation (Figure 3a–d), and the first five eigenvalues exceeded unity, that is, they were relevant. However, this study chose to present the first three, which each explained more than 10% of the variation. In the relationship between the first two factors, there was a closeness pattern among the apparent color, turbidity, total iron, total coliforms, and *Escherichia coli*, and also among the electrical conductivity, total alkalinity, and total hardness (Figure 3b), which reinforces the results obtained from the linear correlations and canonical correlations. In general terms, this pattern was also maintained in the relationships between the first and third factors and between the second and third factors, only with changes in the rotation and positioning of other variables of lesser association.

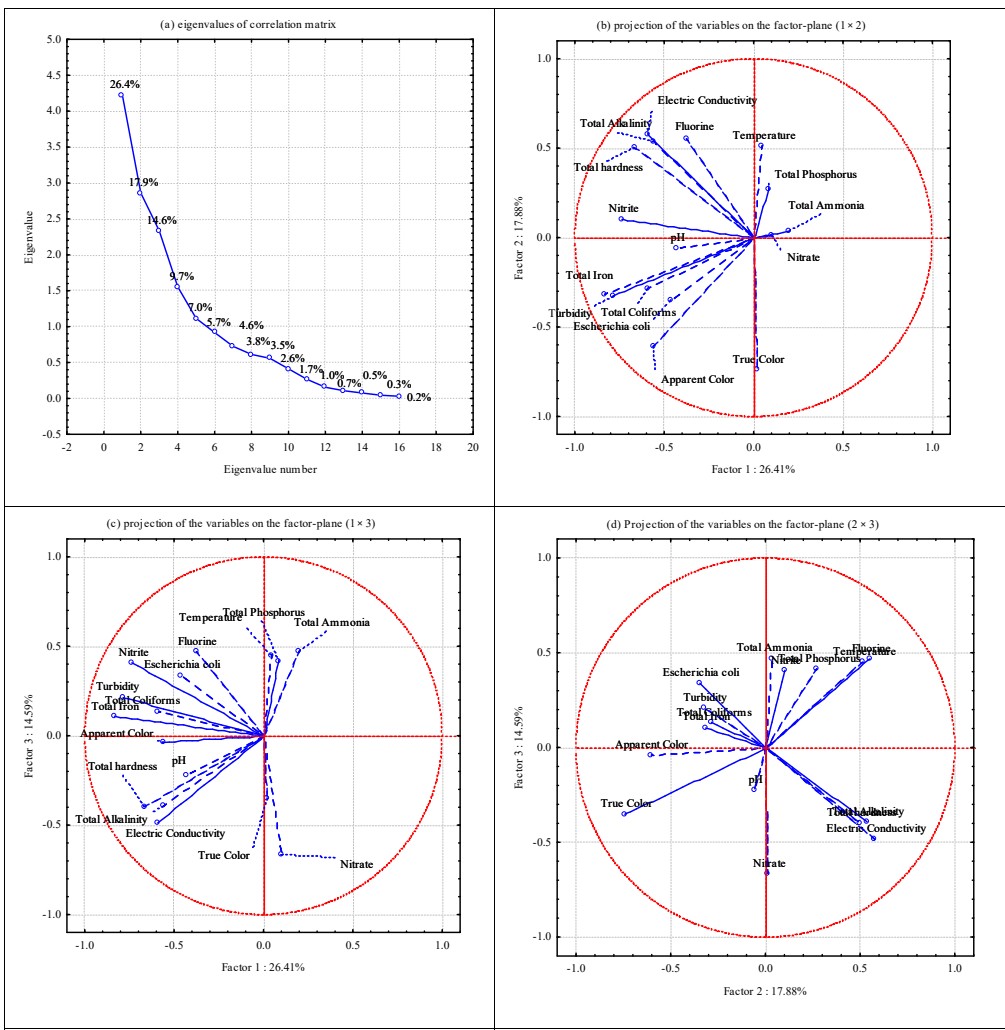

**Figure 3.** Percentage contribution of each principal component (PC), based on its eigenvalue (**a**); graphical representation of the relationship between the first and second (**b**), the first and third (**c**), and second and third (**d**) factor of the PC analysis, based on chemical, physical, specific chemical, and microbiological variables of springs.

## 4. Discussion

The correlation between the pH and TA and between the TA and TH was also observed in the studies by [24]. The TA represents the amount, in mg/L, of $CaCO_3$, thus, the higher the values of this variable in groundwater, the lower the acidity of the water [25].

The TH values indicate the presence of $CaCO_3$ and represent the geological characteristic of the site [26], whereas the values were lower in sandy and clayey terrains, and higher in limestone regions.

The EC variable is related to salinity and indicates the ability of water to transmit an electrical current due to the presence of dissolved substances [27]. Thus, the higher the TA and TH values, the higher the EC value, as the presence of $CaCO_3$ increases the possibility of water transmitting an electrical current [28], and this correlation was also found in [24,25,28].

The positive correlation between the turbidity, color, and TI was also found in [27], due to the color that varies from reddish to black tones in water when Fe is precipitated.

High values of $F^-$ are the result of the interaction between water and rocks and were found in [28]. High fluorine values are a result of the natural interaction between rock and water, while the $NO_3^-N$ and $NO_2^-N$ values are the result of the interaction between water and anthropogenic interferences that occurred in the environment [28], as excessive amounts of $NO_3^-N$ in the soil can cause a reduction in agricultural production. This correlation between $NO_3^-N$ and $F^-$ was also found in [26].

A high correlation between $NO_2^-N$ and TI was also found in [29] and in the present study, this correlation occurred due to the surface and subsurface flow of water. A moderately strong positive correlation was found between the TC and *E. coli* with nitrite in [30].

The principal component analysis was performed in other groundwater quality studies, such as [31,32].

According to [8,9], water quality variables are influenced by several factors, such as the interaction between water and soil, morphological characteristics, erosion and weathering processes, and human influences—especially those related to agricultural activities.

## 5. Conclusions

The total alkalinity, total hardness, and electrical conductivity variables have a strong positive correlation among them, which is similar to what occurs with the turbidity, apparent color, true color, and total iron variables between nitrite and total iron. These correlations found in the present research demonstrate that they occur due to the interaction that water has with the soil and the compounds found there, thus, altering its physical–chemical characteristics. On the other hand, the correlations between turbidity and total coliforms indicate that the location of the springs, their constructive aspects (infrastructure), and sanitary conditions must be taken into account as relevant points that directly influenced the results obtained. The springs do not have masonry protection, they only have plant protection. Moreover, this water is exposed and accessible to animals in these areas. In addition, the result of the surface and sub-surface flows' effects on spring water must be considered.

**Author Contributions:** Investigation: D.S., M.T., A.D.W., G.M.d.R., O.D.P., R.Z. and J.I.G.; conceptualization, methodology, formal analysis, writing—original draft: D.S. and J.I.G.; writing—review and editing: M.T. and J.I.G.; supervision, project administration and funding acquisition: J.I.G. All authors have read and agreed to the published version of the manuscript.

**Funding:** This research received funding from the public agency, Coordenação de Aperfeiçoamento de Pessoal de Nível Superior (CAPES). We would like to thank the agency for providing a research fellowship to D.S.

**Institutional Review Board Statement:** Not applicable.

**Informed Consent Statement:** Not applicable.

**Data Availability Statement:** Not applicable.

**Acknowledgments:** J.I.G. would like to thank all the undergraduate students who are part of the GMA (Environmental Monitoring Research Group of CNPq) and who participated in this study.

**Conflicts of Interest:** The authors declare no conflict of interest.

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
