# Peer review of "Association Patterns among Physical, Chemical and Microbiological Indicators of Springs in Rio Grande do Sul, Brazil"

_water, doi:10.3390/w14193058_

Round 1

Reviewer 1 Report (Previous Reviewer 1)

I already reviewed this paper and the Authors have properly responded to my comments/suggestions. I believe that this manuscript is ready for publication as a "Communication".

Regards

Author Response

Reviewer 2 Report (New Reviewer)

This paper reported by Dr. Débora Seben et al. focuses on association patterns among physical, chemical and microbiological indicators in springs. The topic is interesting. However, there are several major concerns. Firstly, the Introduction is too simple and weakness, it should be strengthened on groundwater quality. Secondly, geographical and hydrogeological conditions in the study area are missing. Thirdly, analysis of association patterns among physical, chemical and microbiological indicators on the basis of canonical correlation analysis is too simple, more multivariate statistical techniques such as hierarchical cluster analysis and principal component analysis could be added.

Specific comments

Abstract

the abstract should be rewritten. Results and conculsions are too few in the present abstract.

Introduction

review on groundwater quality should be strengthened in the Introduction.

Lines 34-36, delete three "on" after land uses.

Materials and Methods

add study area information such as geographical and hydrogeological conditions.

Lines 43-47, add a figure of study area with sampling sites. add sampling details. what about sample storage conditions, especially for microbiological analysis?

Lines 48-54, add analysis methods and the quality controls.

Lines 55-63, add more multivariate statistical techniques such as principal component analysis to strengthen analysis and discussion.

Results

the content should be extended.

Discussions

the content should be extended.

At last, I recommend some references related to groundwater quality and groundwater chemistry explained by principal component analysis may be useful for authors.

A sharp contrasting occurrence of iron-rich groundwater in the Pearl River Delta during the past dozen years (2006-2018): the genesis and mitigation effect. Science of the Total Environment, 829, 154676, 2022.

Natural background levels in groundwater in the Pearl River Delta after the rapid expansion of urbanization: A new pre-selection method. Science of the Total Environment, 813, 151890, 2022.

Spatial distribution and origin of shallow groundwater iodide in a rapidly urbanized delta: A case study of the Pearl River Delta. Journal of Hydrology 585, 124860, 2020.

Groundwater is important for the geochemical cycling of phosphorus in rapidly urbanized areas: a case study in the Pearl River Delta. Environmental Pollution, 260, 114079, 2020.

Distributions and origins of nitrate, nitrite, and ammonium in various aquifers in an urbanized coastal area, south China. Journal of Hydrology 582, 124528, 2020.

Elevated manganese concentrations in shallow groundwater of various aquifers in a rapidly urbanized delta, south China. Science of the Total Environment, 701, 134777, 2020.

Groundwater quality in the Pearl River Delta after the rapid expansion of industrialization and urbanization, Distributions, main impact indicators, and driving force, Journal of Hydrology, 577, 124004, 2019.

Heavy metal(loid)s and organic contaminants in groundwater in the Pearl River Delta that has undergone three decades of urbanization and industrialization: Distributions, sources, and driving forces, Science of the Total Environment 635, 913-925, 2018.

A regional scale investigation on factors controlling the groundwater chemistry of various aquifers in a rapidly urbanized area: A case study of the Pearl River Delta, Science of the Total Environment 625, 510-518, 2018.

Impact of anthropogenic and natural processes on the evolution of groundwater chemistry in a rapidly urbanized coastal area, south China. Science of the Total Environment, 463-464, 209-221, 2013.

Round 2

Reviewer 2 Report (New Reviewer)

This is the second time for me to review this paper. In my opinion, this paper has been improved by authors. However, some concerns are still remained. First, the English level should be improved, I recommend using a professional scientific editing, so the expressions can be tightened a bit. Second, the introduction is still weak, should rewrite. Third, details on materials and methods should be added. Therefore, it needs a moderate revision. 

Specific comments

Introduction

the Introduction is weak, the novelty of this study is still unclear. Authors did not review groundwater qualtiy.

Materials and Methods

Lines 47-48, details on hydrogeological conditions of the study area should be added. It is unacceptable at present form.

Lines 66-67, details on PCA should be added. Do you use tests for PCA? 

Conclusions

Conclusions should be strengthened.

References 22 and 23 are wrong.

Zhang, F., Huang, G., Hou, Q., Liu, C., Zhang, Y., Zhang, Q., 2019. Groundwater quality in the Pearl River Delta after the rapid expansion of industrialization and urbanization: distributions, main impact indicators, and driving forces. J. Hydrol. 577, 124004.

Hou, Q., Zhang, Q., Huang, G., Liu, C., Zhang, Y., 2020. Elevated manganese concentrations in shallow groundwater of various aquifers in a rapidly urbanized delta, South China. Sci. Total Environ. 701, 134777.

Author Response

Response to Reviewer Comments

Manuscript ID water-1850059

Title: Association patterns among physical, chemical and microbiological
indicators of springs in Rio Grande do Sul, Brazil

To the editor and reviewer 2

Thank you very much for your time and the constructive suggestions. We have carefully considered all comments and revised the manuscript accordingly. A list of point-by-point responses is summarized as follows.

Words and phrases written in red and underlined in yellow were corrected by the authors and/or by a professional scientific editing.

Review 2:(yellow and red color in manuscript text)

Comments

This is the second time for me to review this paper. In my opinion, this paper has been improved by authors. However, some concerns are still remained. First, the English level should be improved, I recommend using a professional scientific editing, so the expressions can be tightened a bit. Second, the introduction is still weak, should rewrite. Third, details on materials and methods should be added. Therefore, it needs a moderate revision.

Specific comments

Abstract:

English language corrections by a professional scientific editing; they are presented in red and underlined in yellow color in manuscript text.

In this item, a sentence and conclusion and keywords were added (new lines: 23 to 26 and, key word: 27).

Introduction

English language corrections by a professional scientific editing; they are presented in red and underlined in yellow color in manuscript text.

The Introduction is weak, the novelty of this study is still unclear. Authors did not review groundwater qualtiy.

Response: We appreciate the reviewer’s comments. The Introduction was reorganized and several scientific articles were added to improve the review groundwater qualtiy of the manuscript (new lines: 30-50).

Materials and Methods

English language corrections by a professional scientific editing; they are presented in red and underlined in yellow color in manuscript text.

Lines 47-48, details on hydrogeological conditions of the study area should be added. It is unacceptable at present form.

Response: We appreciate the reviewer’s comments.

a) The text added (new lines: 57-65)

b) new lines 65-68; 73-74; 77-78 and 84-86: with analyzed variables and more details of the collection forms.

Lines 66-67, details on PCA should be added. Do you use tests for PCA? 

Response: We appreciate the reviewer’s comments. In the statistical analysis, principal components analysis (PCA) was added the reference (new line: 85). No tests were performed for PCA analysis, but the upper eigenvalues ​​were discussed one by one.

- new lines: 87-90: Figure 1 was also inserted in the manuscript, with a representative map of the location water collections in the springs.

Results:

English language corrections by a professional scientific editing; they are presented in red and underlined in yellow color in manuscript text.

The figures were numbered in the manuscript being composed of 3 now (Figures 1, 2 and 3).

Discussion:

English language corrections by a professional scientific editing; they are presented in red and underlined in yellow color in manuscript text.

Some references were added, all renumbered and in the text.

Conclusion:

English language corrections by a professional scientific editing; they are presented in red and underlined in yellow color in manuscript text.

Conclusions should be strengthened.

Response: We appreciate the reviewer’s comments. Some sentences were added to improve it, as suggested by the reviewer (new lines: 185-194)

Funding and Acknowledgments:

English language corrections by a professional scientific editing; they are presented in red and underlined in yellow color in manuscript text.

References: Scientific articles written in red and underlined in yellow were were added by the authors. Response: We appreciate the reviewer’s comments. Some references were added, all renumbered and in the text (they are presented in the red and yellow color in manuscript text). Text reference has been withdrawn: - Myers, T. Groundwater management and coal bed methane development in the Powder River Basin of Montana. J. Hydrol. 2009, 368(1–4), 178–193. https://doi.org/10.1016/j.jhydrol.2009.02.001

References 22 and 23 are wrong

These References have been corrected in the text

Zhang, Z.; Huang, G.; Hou, Q.; Liu, C.; Zhang, Y.; Zhang, Q. Groundwater quality in the Pearl River Delta after the rapid expansion of industrialization and urbanization: Distributions, main impact indicators, and driving forces. 2019, 577, J. Hydrol. 124004. https://doi.org/10.1016/j.jhydrol.2019.124004

Hou, Q.; Zhang, Q.; Huang, G.; Chunyan, L.; Zhang, Y. Elevated manganese concentrations in shallow groundwater of various aquifers in a rapidly urbanized delta, south China. 2020, 701, Sci. Total Environ. 134777. https://doi.org/10.1016/j.scitotenv.2019.134777

This manuscript is a resubmission of an earlier submission. The following is a list of the peer review reports and author responses from that submission.

Round 1

Reviewer 1 Report

Line 15: A comma “,” is missing between “electrical conductivity” and “total alkalinity”.

Line 18: What do you mean by strong positive correlation? Please give a number (e.g., correlation > 0.5).

Line 19: What is the physical reason for strong correlation between nitrite and total iron?

Lines 28-30: Please revise.

Line 32-35: Please revise. Also, please add your study area.

Introduction. The authors have failed to organize this section well. This section is too short and does not present the problem statement well. Please rewrite this section.

Line 47: This statement does not need to be supported with references. This is a general statement.

Line 50: What do you mean by condition number? Collinearity is usually diagnosed through variance inflation factor in regression analysis.

Lines 48-55: No references given to support this section. I suggest the authors to see (and cite) “10.1016/j.desal.2010.04.053” and “10.1016/j.eswa.2010.02.020” for canonical correlation analysis and collinearity in the regression analysis, respectively.

Can you please add a short description on QA/QC?

Line 63-64: Please change “N-NO2-” to “NO2-N”.

Figure 1: Can you please determine the statistically significant relations?

Please support Discussion section with more similar works such as “10.1029/2021EA001793” and “10.3390/ijerph15010172”.